# Light Availability Affects the Symbiosis of Sponge Specific Cyanobacteria and the Common Blue Aquarium Sponge (*Lendenfeldia chondrodes*)

**DOI:** 10.3390/ani12101283

**Published:** 2022-05-17

**Authors:** Franziska Curdt, Peter J. Schupp, Sven Rohde

**Affiliations:** 1Department for Environmental Biochemistry, Institute for Chemistry and Biology of the Marine Environment Terramare, Carl-von-Ossietzky University Oldenburg, 26382 Wilhelmshaven, Germany; franziska.curdt@uol.de (F.C.); peter.schupp@uol.de (P.J.S.); 2Helmholtz Institute for Functional Marine Biodiversity, Carl-von-Ossietzky University of Oldenburg, 26129 Oldenburg, Germany

**Keywords:** Porifera, *Synechococcus spongiarum*, symbionts, pigments, aquarium experiment, phototropic, *Lendenfeldia chondrodes*

## Abstract

**Simple Summary:**

Marine sponges contain associated microorganisms in high numbers. This study gives insight into how environmental conditions such as light availability affects the symbiotic relationship between phototrophic bacteria that obtain energy from sunlight for the synthesis of organic compounds and the marine host in which they are found. A controlled aquarium experiment with defined light levels ranging from darkness to higher light intensity demonstrates how light conditions change the number of these bacteria and, consequently, host performance. Sponge growth performance was significantly affected by the number of these bacteria. Moreover, the common blue aquarium sponge *Lendenfeldia chondrodes* has proven to be perfectly suited as a model organism to study marine host-symbiont interactions.

**Abstract:**

Bacterial symbionts in marine sponges play a decisive role in the biological and ecological functioning of their hosts. Although this topic has been the focus of numerous studies, data from experiments under controlled conditions are rare. To analyze the ongoing metabolic processes, we investigated the symbiosis of the sponge specific cyanobacterium *Synechococcus spongiarum* and its sponge host *Lendenfeldia chondrodes* under varying light conditions in a defined aquarium setting for 68 days. Sponge clonal pieces were kept at four different light intensities, ranging from no light to higher intensities that were assumed to trigger light stress. Growth as a measure of host performance and photosynthetic yield as a proxy of symbiont photosynthetic activity were measured throughout the experiment. The lack of light prevented sponge growth and induced the expulsion of all cyanobacteria and related pigments by the end of the experiment. Higher light conditions allowed rapid sponge growth and high cyanobacteria densities. In addition, photosynthetically active radiation above a certain level triggered an increase in cyanobacteria’s lutein levels, a UV absorbing protein, thus protecting itself and the host’s cells from UV radiation damage. Thus, *L. chondrodes* seems to benefit strongly from hosting the cyanbacterium *S. spongiarum* and the relationship should be considered obligatory mutualistic.

## 1. Introduction

Porifera and their associated microbiome are one of the oldest examples of symbioses in marine ecosystems with fossil records dating back to the Cambrian explosion around 542 million years ago or even before that [1,2]. Nowadays, an estimated 15,000 different sponge species inhabit the world’s oceans from deep sea to shallow waters and throughout all climate zones [3]. Sponges play an important role as one of the drivers of the ecosystem functioning in extremely nutrient poor waters such as coral reefs. Via the sponge loop, they convert dissolved organic matter (DOM) into particulate matter that becomes accessible to detrivores [4]. In this process, the bacterial symbionts play a decisive role as they confer supplemental nutrition, e.g., via photosynthesis. In addition, symbionts were shown to supply secondary metabolites which can serve the host as chemical defense or protection, e.g., antifeedants, UV filters, or antibiotics. In turn, the symbionts receive host metabolites as well as shelter [5].

In recent years, many studies have described the presence of distinct microbial communities in sponges. 16S rRNA analyses have shown that there are numerous sponge specific bacteria clusters, that are found exclusively inside certain sponges. Even unrelated sponges from distant geographical location have partly overlapping microbiomes. This leads to the suggestion that these symbionts fulfill important biological and ecological functions that are crucial for the survival of the host [6,7,8]. Depending on the amount and diversity of bacteria inside their tissue, sponges are categorized as low microbial abundance (LMA) with $10^{5}$ to $10^{6}$ bacteria per gram or high microbial abundance (HMA) sponges with $10^{8}$ to $10^{10}$ microorganisms per gram sponge tissue (more than 25% of the area) [9,10,11].

However, the exact functions of bacterial symbionts remain mostly unknown, even though important microbial processes have been identified that include the carbon and nitrogen cycling and likely an important role in the sulfur and vitamin metabolism [12,13]. In addition, there are many secondary metabolites produced by the symbionts which might benefit the sponge, e.g., as chemical defense against predators and antibiotics against pathogens [7,14]. Bacterial symbionts can be divided into specialists interacting only with selected sponge species, generalists interacting with a broad range of species, and opportunists which can occur in many species, but less regularly and without benefits to the host [15]. A microbiome survey has shown the exceptional microbial diversity, but also that the core sponge microbiome remains stable within species with a clear tendency towards generalists and specialists as opposed to opportunists [8]. Still, the variability and functions of sponge symbionts are mostly unclear, as well as the conditions which favor symbiont abundance and metabolic activity. There are only very few ecophysiological studies in suitable model sponges due to the difficulty of successfully cultivating sponges in aquaria [16], but see [17,18]. The group of phototrophic cyanobacteria are well suited to investigate host-symbiont beneficial interactions since the metabolism of cyanobacteria can easily be altered by different light conditions. In this context, *Lendenfeldia chondrodes* [19] and its cyanobacterium *Synechococcus spongiarum* represent an excellent model system since *L. chondrodes* seems to rely heavily on photosynthetic products. The abundance of cyanobacteria was found to vary with light availability: *L. chondrodes* on reef sites with low light availability and less symbionts have been reported from Palau [20]. *L. chondrodes* is a spiculeless sponge which can be found in various morphological forms: Encrusting, foliose and cup shaped. Its tissue is smooth and the surface shiny with mostly very small, unobtrusive oscula. Its coloring can be blue, brownish, purple, or green [21]. The individual in this study was purple, but some samples changed coloration at different light conditions to grey and later white. The core bacterium of *L. chondrodes* is the yet uncultivated, rod shaped cyanobacterium *S. spongiarum*. It has a length of 1 to 2 µm and a thickness of 0.5 to 1 µm [18]. *S. spongiarum* has been reported in more than 40 sponge species from very different geographical locations such as Mediterranean, Red Sea, Zanzibar, the Caribbean, tropical and temperate Australia, Bahamas, and Guam [22,23]. Generalist symbiont groups of Candidatus *S. spongiarum* have been found in unrelated host sponge species, but there are also host specific groups of this bacterium [24].

In this study we applied different light conditions to investigate the nature of the symbiosis of the sponge *L. chondrodes* and the cyanobacteria *S. spongiarum*. Changes in symbiont abundance, pigment concentrations, photosynthetic performance, and their effect on sponge growth were analyzed to reveal the physiological processes within this symbiotic relationship.

## 2. Materials and Methods

### 2.1. Aquarium Experiment Design

Specimens of *L. chondrodes* were cultivated at the aquarium of the ICBM in Wilhelmshaven. One large sponge individual was cut into similar sized pieces of approximately 8 cm^2^ with a thickness of approximately 5 mm. Pieces were fixed on tiles of 22 cm^2^ with the help of cable ties. Five to seven days after attachment of the sponges, cable ties were removed. Sponge pieces were placed into aquaria of 1.2 L with independent water supply. After one week of adaptation to the new environment, light conditions were manipulated. Aquarium lamps (Radion G4, EcoTech Marine, Bethlehem, PA, USA) with adjustable light intensities illuminated the samples in a 12 h/12 h day/night cycle. We used 4 aquaria per light condition. Eight individuals (2 per aquarium) were exposed to higher light of 100 µmol photons/m²s (400% of control), eight to control light condition at 25 µmol/m²s (control, 100%), eight to dimmed light at 12 µmol/m²s (50% of control) and eight were kept in complete darkness (0%). The applied light conditions of 400% and 100% are commonly found on turbid reefs at water depths from 4–10 m [25,26]. Higher light conditions appeared to stress *L. chondrodes* with signs of bleaching and growth reduction (unpublished data). Photosynthetically active radiation (PAR) was measured 5 cm below the water surface with a PAR meter (apogee, Colorado Springs, CO, USA). To remove all light sources for the dark condition, the aquaria were painted black with several layers of liquid pond liner (Tripond, Mainhausen, Germany) in advance. The darkened aquaria were dried and rinsed thoroughly for five days to prevent contamination with solvents or other chemical compounds. To keep light to a minimum, black plastic lids were placed on top of the aquaria and water supply was established with black hoses. One sponge sample in the 400% light condition died after a few days. The experiment was run for 67 days.

### 2.2. Photosynthetic Rates

After twelve hours of darkness, the optimum quantum yield (*y*) of each dark-adapted sponge piece was measured with a Diving-PAM fluorometer (Walz, Effeltrich, Germany). The PAM fluorometer excites chlorophyll *a* fluorescence by pulse modulated red light (LED, 655 nm) with pulse-width of 3 µs. The LED-light is passed through a cut-off filter resulting in an excitation band peaking at 650 nm, with a very small fraction at wavelengths beyond 700 nm. Fluorescence is detected at wavelengths longer than 700 nm, as defined by a long-pass filter. The photosynthetic yield *y* is defined as $y=\left(F_{m}-F\right)/F_{m}$, where *F* is the minimal fluorescence before and $F_{m}$ the maximal fluorescence during the saturation pulse, see [27]. Signals of five different surface areas of each sponge piece were recorded from a distance of one centimeter from the sponge’s surface. The yield values were calculated as means of repetitive measurements at different sponge surface areas.

### 2.3. Growth

All sponge pieces were photo-documented at the beginning and at seven time points thereafter. The encrusting growth form of most of the samples made it possible to use area as a measure for sponge biomass increase (growth). Sponge area was measured with the polygon selection tool in Fiji ImageJ [28]. Growth was calculated as percent area increase with respect to day 0.

### 2.4. Cyanobacteria Density

To estimate cyanobacteria densities at the end of the experiment, pieces of approximately 30 mm times 5 mm edge length were removed from each sponge for fluorescence microscopy. Pieces were taken from the central part of the sponge rather than from the outer freshly grown tissue. Samples were fixated in 4% paraformaldehyde (PFA) diluted in phosphate-buffered saline (PBS) for 24 h. Fixated samples were washed three times in PBS and placed into 30% sucrose for cryoprotection. Samples were stored in sucrose solution at 4 °C for 24 h. For the preparation of cryosections, samples were placed onto a 2 mm plateau of tissue freezing medium (TFM) on a cryoholder and embedded into TFM with the help of a cylindrical mold made of aluminum foil inside the cryotome (Leica, Mannheim, Germany) at −20 °C. Orientation of the sponge cube on the holder allowed longitudinal cuts to be made. After freezing, cylindrical blocks of 5 to 10 mm in diameter were trimmed into rectangular cuboid shape with a scalpel. Sections of 10 µm thickness were cut at −20 °C. Cryosections were placed on adhesive glass slides (Superfrost, Thermofisher, Waltham, MA, USA). After sectioning, the glass slides with the tissue sections were dried at 60 °C for several hours to properly adhere to the glass surface and stored at −20 °C until further processing. Crysections were washed three times with PBS for 15 min, air dried, and embedded with 15 μL embedding medium (Roti-Histokitt, Carl Roth, Karlsruhe, Germany). Samples were left to dry for several hours before they were imaged on a Zeiss Axiophot at 1000× magnification. Objective lens was an oil immersion plan apochromat 100× with a numerical aperture of 1.3. Ten images in different areas of each of the 31 samples were taken with an integrated camera. One field of view had the size of 65 µm × 87 µm. The numbers of cyanobacteria were estimated using the mask tool in Fiji ImageJ to estimate the fluorescent area per field of view.

### 2.5. Pigment Concentration

The remaining sponge tissues (1–2 g wet weight) were used for pigment extraction. Samples were ground in an homogenizer for one to two minutes at 9500 rpm and placed into glass vials with 10 mL MilliQ water. Samples were kept in the dark to avoid degradation by light. After 90 s of sonication, they were subjected to three freeze-thaw cycles (−20 °C, 6 h/room temperature, 4 h). After the last thaw cycle, samples were centrifuged. 600 μL of the supernatant from each sample was pipetted onto a 96 microwell plate (200 μL per well, 3 replicates). The absorbance over the entire visible spectrum (300–900 nm) was measured in a microplate reader (Synergy 1, Biotek Instruments, Winooski, VT, USA). Phycocyanin and phycoerythrin concentrations ($c_{PC}$ and $c_{PE}$) in aqueous extracts were calculated as follows [29]:(1)$$c_{PC}=0.12\times \left[\left(A_{564}-A_{592}\right)-0.20\times \left(A_{455}-A_{592}\right)\right]$$
(2)$$c_{PE}=0.15\times \left[\left(A_{618}-A_{645}\right)-0.15\times \left(A_{592}-A_{645}\right)\right]$$
where $A_{\lambda }$ is the absorption coefficient at the respective wavelength λ in nanometers. After removal of the aqueous extract, samples were freeze dried and 10 mL of acetone was added. After 18 h at 4 °C, the samples were centrifuged again, and the supernatants were measured in the microplate reader. Chlorophyll *a* concentration, $c_{chla}$, was calculated as follows [30]:(3)$$c_{chla}=11.85\times \left(A_{664}-A_{750}\right)-1.54\times \left(A_{647}-A_{750}\right)-0.08\times \left(A_{630}-A_{750}\right)$$
and carotenoid concentration, $c_{caro}$, (lutein/zeaxanthin) with [31]:(4)$$c_{caro}=\frac{1000\times \left(A_{450}-A_{750}\right)+411.307\times \left(A_{633}-A_{750}\right)-1822.8522\times \left(A_{647}-A_{750}\right)}{198}$$
chlorophyll *a* and the carotenoids lutein and or zeaxanthin were identified by mass spectrometry. Since lutein and zeaxanthin have exactly the same mass, they could not be differentiated further. The pigment concentrations above are given in units of mg/L. The respective concentration per wet weight sponge, $c_{wetweight}$, in the extraction volume *V* for wet weight $m_{wet}$ was calculated as:(5)$$c_{wetweight}=c\times V/m_{wet}$$

### 2.6. Data Analysis

All data sets: Growth, yield, pigment concentrations, and cyanobacteria density (8 replicates each, 7 replicates for the 400% light samples) were checked for normal distribution with the Kolmogorow–Smirnow Test with the MATLAB function ‘kstest()’. ‘kstest()’ was applied to the entire data set after transformation to normal distribution with the equation:(6)$$x_{0}=\left(x-\bar{x}\right)/s_{x}$$
where $x_{0}$ is the transformed value, *x* the value of each sample and $\bar{x}$ and $s_{x}$ the mean and standard deviation of all data. For the yield values, *y*, numbers were transformed to $y_{0}=arcsine\left(y\right)$ to meet the requirements for ANOVA [32]. One-way ANOVA was carried out with MATLAB using the functions ‘anova1’ with ‘light intensity’ as nominal variable. Tukey’s test was used as post hoc test. For two data sets (yield values of day 56 and day 67) the hypothesis of normal distribution was rejected by the Kolmogorow–Smirnow Test. ANOVA was applied nevertheless, since there is sufficient evidence from recent studies that the one-factorial ANOVA is robust to a violation of normal distribution, especially if the size of the groups is very similar [33,34,35,36]. Data was plotted employing the Python data visualization library seaborn [37].

## 3. Results

### 3.1. Photosynthetic Rates

The photosynthetic yield (*y*) showed a strong decline for the samples that were kept in the dark (see Figure 1). In the other light treatments, there were only minor changes. Yields for all treatments were similar on day 0 (ANOVA p=0.72) with yield values between y=0.52 and 0.53. After 21 days, the yield of the darkened samples started to drop to y=0.41 and after 36 days to y=0.36. After 56 days, the yield values for the samples at dark were below the detection limit of the fluorometer. For all other treatments, the yield values remained on a similar level between y=0.45−0.57 over the entire duration of the experiment. ANOVA showed, except for the data from day 0, that the mean yield differed significantly. Multicomparison test (Tukey’s) revealed significant differences when sampled at day 21, day 36, day 56 and day 67. Furthermore, at day 21 the yield in the 50% light treatment was significantly higher than the yield of the 400% light samples (p=0.003).

### 3.2. Growth

Growth varied between the different light conditions (see Figure 2). The low and no light-samples (0% and 50% light) grew less than 15% with respect to day 0, over the entire experiment. At 50% light, sponge growth was slightly higher than that for the dark condition. The sponges kept at 0% light stopped growing after 30 days and started to loose tissue. Growth was highest for the samples in the the 100% and 400% light conditions. At the end of the experiment, growth for the 100% and 400% light conditions was 79% and 106% respectively.

On day 15, the sponge growth of the 400% light samples and 0% light samples were significantly smaller from control (p<0.012). On day 22, this changed: The 0% light and the 50% light samples’ growth was significantly smaller than control (p<0.001). This difference became larger on day 29 (p<0.0003) and on day 43 (p<0.00002) and remained similar on day 50 (p<0.00008), day 57 (p<0.000006), and day 67 (p<0.000001). Although slightly higher after day 50, the growth of the 400% light samples did not differ significantly from the control at any time point. The growth of the 0% and 50% samples also did not differ from each other significantly at any time point.

### 3.3. Pigment Concentrations

Mean pigment concentrations (phycocyanin, phycoerythrin, chlorophyll *a* and carotenoids) were analyzed at the end of the experiment. Pigment concentrations decreased significantly for the sponges kept in the dark. The concentrations for all pigments dropped to nearly zero (see Figure 3) and were significantly lower than the concentrations in the 100% and 400% light treatments. The phycocyanin and phycoerythrin concentrations at 100% and 400% light were significantly enhanced with respect to 0% light samples (ANOVA, p<0.0002 for phycocyanin, and p<0.0011 for phycoerythrin). The phycocyanin concentration in the 50% light samples was significantly higher than the 0% light samples as well (ANOVA, p<0.006), but phycoerythrin concentrations did not differ significantly between the 0% and 50% light samples. Pigment concentrations of sponges kept at 50%, 100% and 400% light did not differ significantly and ranged between 0.07–0.09 µg/g phycocyanin, 0–0.02 µg/g phycoerythrin, and 44–59 µg/g chlorophyll *a*. Only the carotenoid concentrations differed significantly with higher concentrations at 400% light (30 µg/g) compared to the 50% and 100% light conditions (14 µg/g), (ANOVA, $p<5.12\times 10^{-9}$ and $p<4.35\times 10^{-9}$, respectively).

### 3.4. Cyanobacteria Density

Evaluation of fluorescent microscopy images showed that cyanobacteria were very abundant in the sponge samples kept under different light conditions at the end of the experiment and were distributed throughout the sponge’s tissue except in the 0% light samples where they were absent. Besides this difference, mean cyanobacteria densities did not differ significantly and ranged between 96,000–140,000 cyanobacteria per mm^2^ for the different light conditions (see Figure 4). Examples of fluorescent images and exemplary photos for each treatment are shown in Figure 5.

### 3.5. Linear Correlation Evaluation

The linear correlation between yield, cyanobacteria density, and pigment concentrations versus sponge growth were calculated using the Python library NumPy [38]. Photosynthetic yield correlated significantly with sponge growth (r=0.488; p<0.0054). Similarly, cyanobacteria density correlated with sponge growth (r=0.517; p<0.003), see Figure 6a,b. There was a stronger linear correlation between the phycobilins and growth (r=0.624, p<0.0002 for phycoerythrin and r=0.624, p<0.0002 for phycocyanin). Similarly, lutein/zeaxanthin level correlation was at r=0.621 (p<0.0002), see Figure 7c,d. The linear correlation between chlorophyll *a* concentration and growth was lower, but still significant (r=0.42; p<0.02), see Figure 7a.

## 4. Discussion

This study demonstrates that the cyanobacterium *S. spongiarum* and the marine sponge *L. chondrodes* live in an intimate symbiosis. *L. chondrodes* is a fast growing and robust species and therefore a well-suited organism to study sponge related symbioses as shown in a previous study [18]. Our results strongly suggest that *L. chondrodes* relies obligatory on the presence of a phototrophic sponge specific cyanobacteria and thereby on light availability. In the absence of light, sponges bleached due to symbiont loss, lost all pigmentation, stopped growing and consequently died.

The here observed sponge-symbiont association demonstrates that the interactions within a sponge holobiont can be obligate with *L. chondrodes* relying on survival on its cyanobacterial symbiont. However, a study focusing on the role of *S. spongiarum* in the Caribbean sponge *Xestospongia exigua* did not observe a significant effect of shading on the growth of the sponge after two weeks but a significant decline in chlorophyll *a* concentration [39]. The absence of tissue loss was discussed as the result of a facultative relation of *X. exigua* and *S. spongiarum*. Authors argued that S. spongiarum might exploit resources of the hosts sponge without significantly affecting sponge biomass (being a commensal symbiont) and during shading Synechococcus symbionts may be consumed by its hosts or may disperse, explaining the reduction in symbiont numbers [39]. These contrasting results with the same symbiont species could indicate that *S. spongiarum* during symbiosis with *L. chondrodes* behaves rather like an obligate symbiont (host specificity of the symbiosis). Alternatively, different experimental conditions could have contributed to the observed results in Thacker’s study. The reduced light conditions in the study of Thacker et al. were applied for only two weeks. The generally low growth rates of *X. exigua* [39] could have prevented the detection of growth differences between light exposed and shaded sponges, while reductions in cyanobacteria were already significant in the same period of time. Longer shading periods could have revealed sponge growth reduction indicating an obligate relationship between *X. exigua* and *S. spongiarum*. In our study, host specific traits of *L. chondrodes* could have caused the high tolerance of *S. spongiarum* to reduced light conditions. The loss of cyanobacteria was detectable only in the complete darkness treatments not in shaded conditions, indicating a higher tolerance of *S. spongiarum* to shading within *L. chondrodes* than in *X. exigua*. As a consequence, one might argue that the relation of sponges and their symbionts is rather controlled by host specific traits than by the species identity of the symbiont. However, this would not be in accordance with the hypothesis that a generalist symbiont like *S. spongiarum*, which is found in many sponge species, has a facultative relation to its host, while specialist symbionts are living obligatorily in their hosts [39].

The ecological interpretation of our results needs to be treated with caution since we used clonal sponge samples in our experiment. Any genetic population variability is therefore lacking, which may have affected the outcomes in our observations. However, our results demonstrate clearly that the symbiosis of the sponge *L. chondrodes* and *S. spongiarum* allows an adaptation to a large range of light conditions. Studies on other sponge species have shown that sponge responses to shading experiments are sponge specific and differ along a wide range of light conditions. While species such as *Lamellodysidea chlorea* and *Chondrilla nucula* are other examples of an obligate mutualism with their phototrophic symbionts, which do not survive experimental shading [39,40], shading only reduced growth rates of the sponges *A. aerophoba* and *A. fulva*, indicating a facultative mutualism [41,42]. Other species like *X. muta*, *Neopetrosia exigua* and *N. subtriangularis* were not affected by shading indicating a commensalistic symbiosis [39,41,43,44]. Even though *L. chondrodes* relies on *S. spongiarum* in order to survive, the bleached sponges can remain in a starvation stage for four to five weeks of complete light deprivation before the loss in cyanobacteria abundance starts to show effects. To compensate for the loss of organic matter from phototrophic symbionts, sponges can either ingest particulate organic carbon (POM) or dissolved organic matter (DOM). POM uptake is facilitated by pumping water and phagocytosis in the chonacyte chambers [45,46,47] and the uptake rates depend on a wide variety of factors that vary between sponge species [48]. DOM uptake of sponges has gathered renewed attention since it has been shown that uptake rate can even be higher than the highest recorded microbial plankton uptake rates and DOM accounted for over 90% of the sponge diet [49]. To our knowledge, there is no information available on heterotrophic feeding or DOM uptake rates of *L. chondrodes*. Further studies need to address whether nutrition in this period is based on filtration, DOM uptake or the use of stored compounds. It is also unknown whether *S. spongiarum* can repopulate completely bleached specimens of *L. chondrodes*. There is evidence of a vertical transmission (from parent to offspring) of the cyanobacteria *S. spongiarum* in the sponge *Chondrilla australiensis* [50], but the widespread distribution and common occurrence of *S. spongiarum* within many sponge species suggest also a horizontal transmission through a free-living stage of the symbiont. Horizontal transmission would allow the recovery of *L. chondrodes* after bleaching events.

Our results show a considerable growth reduction of *L. chondrodes* with decreasing light conditions and a complete lack of growth in darkness. A similar effect was observed by S. Vargas in preliminary experiments with several individuals of L. chondrodes (personal communication). *L. chondrodes* is reported to be found in well-lit shallow tropical waters [20], where average benthic irradiation of up to 500 µmol/m²s are found [51]. At these natural well illuminated conditions, *L. chondrodes* contained chlorophyll *a* concentrations of 300–700 μg sponge dry weight. This is significantly less than our findings of approximately 2000 μg sponge dry weight under illuminated conditions. (Dry weight was 1.6–3% of wet weight). Freeman et al. (2016) [20] attributed the high chlorophyll *a* concentration rather to high symbiont numbers in the sponge tissue than a cyanobacteria adaptation by increased chlorophyll *a* concentrations. Our study demonstrated relatively stable cyanobacteria and chlorophyll *a* concentrations at illuminated conditions, therefore we cannot rigorously verify this assumption. However, the similar chlorophyll concentrations compared to increasing carotenoid concentrations with increasing light indicate a relatively stable chlorophyll *a* concentration within cells. Interestingly, our experiments revealed no differences in photosynthetic yield light intensities greater than 12 μmol/m²s. This could indicate that photosynthesis is light saturated at relatively low light conditions, which correspond roughly to natural light conditions found at depths greater than 10m depth or in shaded areas [51]. This saturation effect of photosynthetic yield was mirrored also by the bacterial densities and the related pigments chlorophyll *a*, phycocyanin, phycoerythrin. The bacteria densities did not correlate with the light intensities or growth but remained relatively stable in all but the light deprived conditions. However, the saturation effect is not supported by our sponge growth data, since sponges grew more at 400% light than at 100% light. At 50% light, the sponges did not grow significantly, even though the photosynthetic yield was kept at similar levels compared to the high light levels. We can hypothesize that at 50% light the physiological energy was barely sufficient to preserve the photosynthetic yield and the abundance of cyanobacteria, but did not allow any allocation to sponge growth. At higher light levels, the higher energy budget allowed the maintenance of bacterial numbers and photosynthetic yield and the additional investment in sponge growth. Although the benefit of the symbionts must be smaller at lower PAR, there is no indication that the sponge can regulate numbers of *S. spongiarum* nor that the sponge would benefit from it. A study that investigated phototrophic sponge metabolic response on sedimentation found that some species increase their respiration rate to compensate for lower phototrophy [52]. Our findings suggest that *L. chondrodes* is able to switch to some degree to heterotrophy in reduced light conditions. This mechanism was also found in the species *Petrosia ficiformis*, *Aplysina fulva* and *Neopetrosia subtriangularis* [40,41].

Furthermore, our study confirms the results of previous field studies [20] in terms of effects of shading and lack of light to cyanobacterial symbionts. Likely, the host sponge produces chemical compounds that are beneficial, or even vital, to the symbiotic bacteria.

So far, we do not know whether at long term light deprivation, the cyanobacteria die passively, leave the sponge, or are actively ingested by the host to compensate for the loss of photosynthetic products. The photosynthetic rates, estimated by the photosynthetic yield, increased at higher PAR after some adaptation period. However, correlations between yield and growth were rather low, which might be related to the methodology used. Cyanobacteria are characterized by a high content of phycobilins and carotenoids. The PAM-fluorometer measures chlorophyll *a* related activity only, but not the accessory phycobilins, or the carotenoids, which showed a higher linear correlation to sponge growth and might have masked photoadaptations that are commonly analysed by chlorophyll *a* content. Although chlorophyll *a* is the main pigment of photosynthesis, it cannot absorb light in the range of 450 to 550 nm. Especially in the water, where red light is strongly absorbed the accessory pigments play an important role in making light in the blue/green spectrum available for photosynthesis. The antioxidant and UV-absorbing carotene levels increased at 400% light conditions. This reaction serves a photoprotective function and is known from free living clades of genus *Synechococcus* [53,54]. This might explain the reduced photosynthetic yield in the beginning of the experiment (day 21) where high light intensities might have reduced the yield compare to the sponges at 50% light. Photoprotective carotenoids might not have been sufficiently produced yet. Increased lutein/zeaxanthin levels at the end of the experiment correlated strongly with sponge growth and allowed the adaption to higher PARs.

The study showed that *L. chondrodes* needs a minimum PAR of more than 12 µmol photons/m²s to grow, it can survive lower light levels, but stops growing. Further studies need to address whether the phenotypic variability found in this study is characteristically for different populations of this species. Experiments including the genetic population variation could give insights whether adaptations of symbionts and hosts to different light ranges are more phenotypic or genotypic responses. Also, assays investigating carbon sources and flows could identify how resources are allocated within the symbiosis in more detail.

## Figures and Tables

**Figure 1 animals-12-01283-f001:**
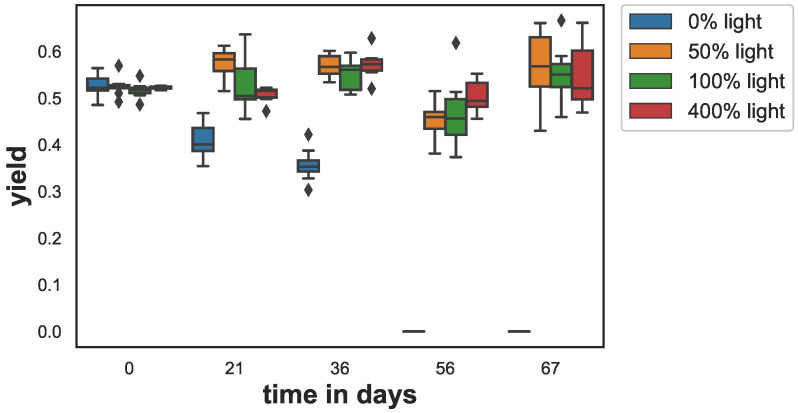
Trend of the fluorescent optimum quantum yield as a proxy for photosynthetic activity. The box shows the quartiles of the dataset while the whiskers show the rest of the distribution, except for outliers that are plotted as diamonds.

**Figure 2 animals-12-01283-f002:**
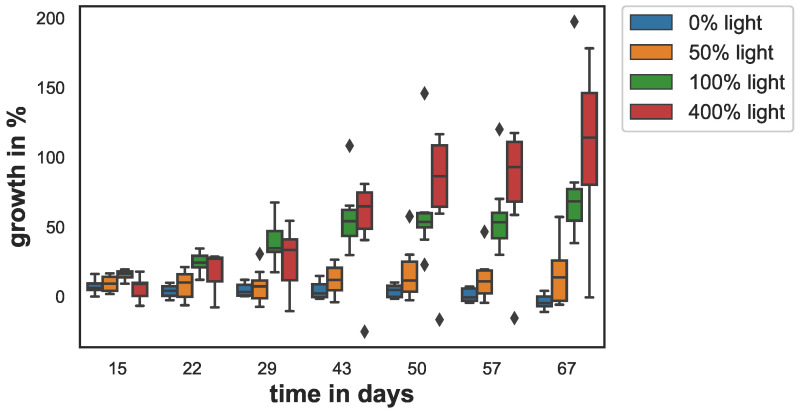
Sponge growth in percent area increase with respect to initial area over the course of the experiment. The box shows the quartiles of the dataset while the whiskers show the rest of the distribution, except for outliers that are plotted as diamonds.

**Figure 3 animals-12-01283-f003:**
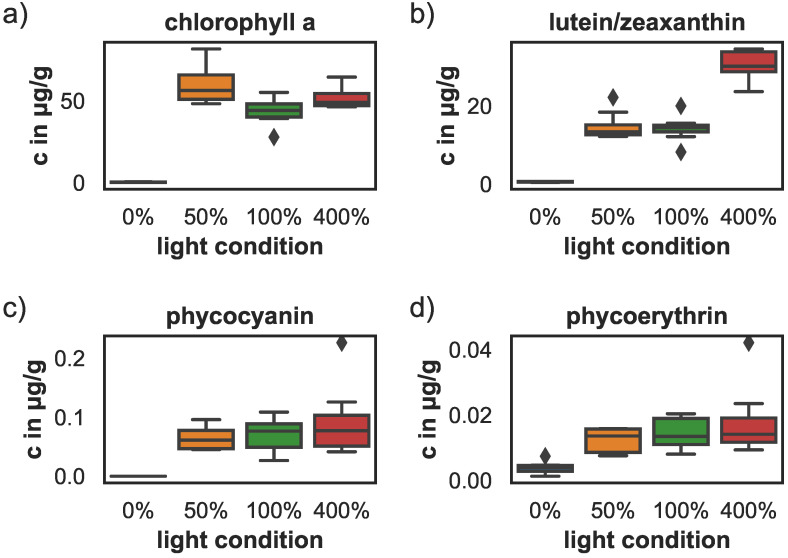
Pigment concentrations after 68 days under varying light conditions. (**a**) chlorophyll *a* concentrations, (**b**) carotenoid concentrations, (**c**) phycocyanin concentrations, (**d**) phycoerythrin concentrations. The box shows the quartiles of the data-set while the whiskers show the rest of the distribution, except for outliers that are plotted as diamonds.

**Figure 4 animals-12-01283-f004:**
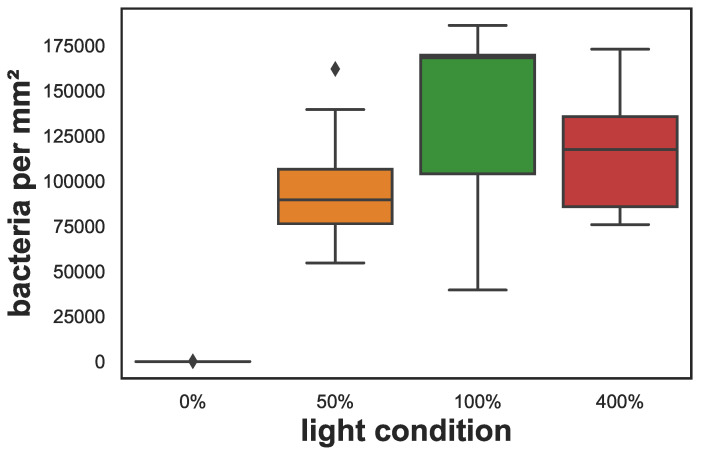
Cyanobacteria density in cryosections of the sponges after 67 days of exposure to different light conditions. The box shows the quartiles of the data-set while the whiskers show the rest of the distribution, except for outliers that are plotted as diamonds.

**Figure 5 animals-12-01283-f005:**
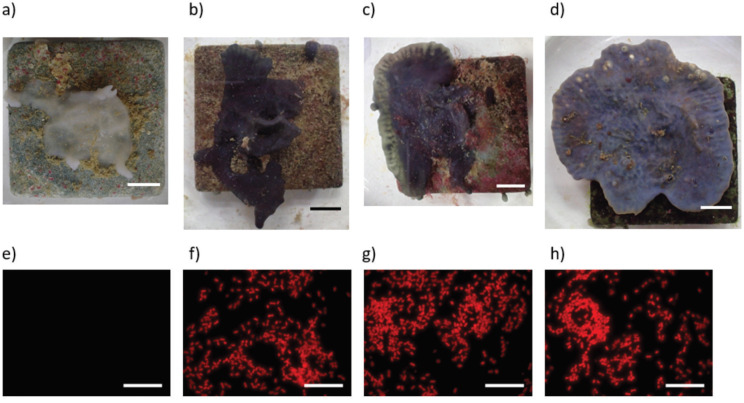
Exemplary photos of the sponge’s appearances in the different light conditions after 68 days. (**a**) darkened, (**b**) 50% light, (**c**) 100% light, (**d**) 400% light. Fluorescence Microscopy images show the density of autofluorescent cyanobacteria (**e**) darkened, (**f**) 50% light, (**g**) 100% light, (**h**) 400% light. Scale bars 10 mm (**a**–**d**) and 20 µm (**a**–**d**).

**Figure 6 animals-12-01283-f006:**
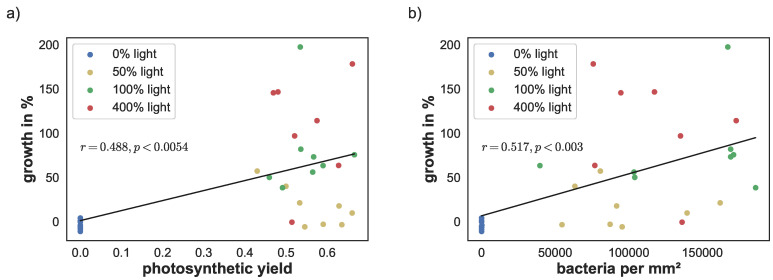
(**a**) Correlation between photosynthetic yield on day 67 and sponge growth and (**b**) correlation between cyanobacteria density and sponge growth.

**Figure 7 animals-12-01283-f007:**
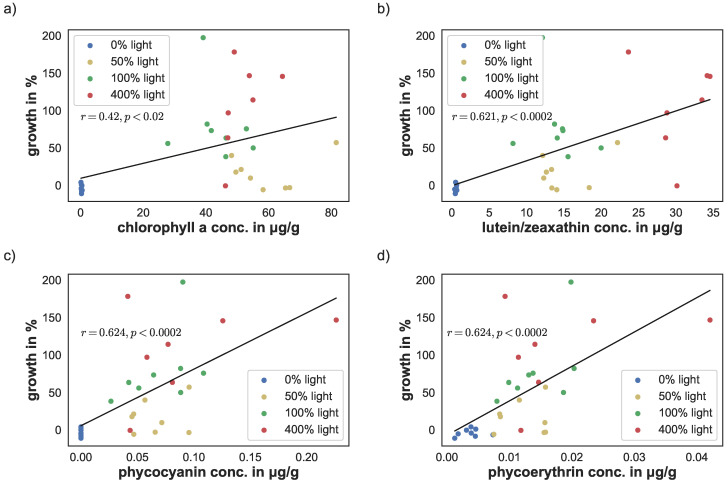
Correlation of pigment concentration and sponge growth for (**a**) chlorophyll *a*, (**b**) lutein/zeaxanthin, (**c**) phycocyanin, and (**d**) phycoerythrin.

## Data Availability

Data available upon request.

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
