# Peer review of "Light Availability Affects the Symbiosis of Sponge Specific Cyanobacteria and the Common Blue Aquarium Sponge (Lendenfeldia chondrodes)"

_animals, 2022, doi:10.3390/ani12101283_

Round 1

Reviewer 1 Report

The manuscript from Curdt et al. entitled: “How light condition affects the symbiosis of sponge specific cyanobacteria and the Common Blue Aquarium Sponge (Lendenfeldia chondrodes)” exposes the results from a controlled aquarium experiment using the photosynthetic sponge Lendenfeldia chondrodes, under light exposures ranging from darkness to moderate light intensity, and for a period of 67 days. In this study the major focus was to assess the status of the symbiotic interaction between the sponge host and its associated cyanobacteria (cyanobacterium Synechococcus spongiarum), by monitoring in several time points (7) physiological parameters such as photosynthesis yields, pigments concentrations, cyanobacteria cell densities per weight of sponge tissue, and host growth rates measured by area increase. The authors report misleading effects in samples kept in dark conditions with respect to the rest of the light treatments in relation to all parameters. They also observe some particularities including carotenoids scalation in samples exposed to the intense light treatment, reduced growth of dark and 50 % light exposed groups, increased photosynthetic yield in 50% light samples in the initial time points.

In general, the topic is not new, and the outcomes obtained do not bring any major novelty to the field, as it is from long known that light restrictions negatively affect photosymbiotic partnerships. The observed effects reported are similar to those published in many old and new works: normally entailing losses of photosymbionts, pigments, and often consequent reduction of host growth and fitness. One major concern in the experimental design is the fact of using a single large individual divided by clones to represent all replicates across treatment groups to run all the experiments. Note that with this scheme you are not catching population genetic variability in your results and potential diverse responses, thus you do not really have replicability for answering ecological responses, and this should be addressed in the text and in particular the Discussion.

Still with these flaws, I think this work can be of interest if focused in a certain manner.

So I invite the authors deeply improve their work by addressing the points below and in the enclosed marked PDF document with marks and comments.

The title is strongly recommended to be improved and made more soundly. Avoid starting with “How …” for instance.

The text is more or less well written and clear until the Discussion, yet it has many points and inquiries raised in the marked PDF attached to this review (please check it!).

Some points in the methods need more detail and clarifications. Authors could think of adding a scheme drawing of the experimental setting.

The Discussion needs to be deeply edited and re-written to address specifically the experimental outcomes, and provide physiological interpretations to the actual data: So “what is happening?”. Authors should very carefully handle the fact of having a single individual, hence no population genetic variability to build any realistic ecological hypothesis. So, as said, they should stick to discussing well their data and build a story on their specific outcomes. Also they need to add a conclusive sentence and some final lines of future plans of innovation or improvement to complement what is known in the topic.

Please, consult all comments, suggestions and inquiries in the enclosed marked PDF.

Reviewer 2 Report

The manuscript examines the effect of light on symbiotic cyanobacteria and sponge hosts. The authors found light intensity is positively correlated with the growth of symbionts and hosts.  They also found an increase of protective pigments in response to high irradiation. The study provides a baseline to further develop aquarium sponge to study host-microbiome interactions. 

Minor issues: 

  1. Are there significant difference between treatments. Statistical analysis is missing for Figure 1-4
  2. Correlation plots  for Figure 6 and 7 are largely determined by samples in the dark. I suggest authors remove dark samples to see if light intensity shows a dose dependent effect on growth. 

Reviewer 3 Report

Manuscript Number: animals-1614927

The manuscript entitled “How light condition affects the symbiosis of sponge specific cyanobacteria and the Common Blue Aquarium Sponge (Lendenfeldia chondrodes)” provides novelty information about the effects of different light conditions in symbiosis of cyanobacteria Synechococcus spongiarum and sponge Lendenfeldia chondrodes. The work presents the originality of the results and the relevance of the subject according to the matter of “Animals” journal. In general, the manuscript is well written and structured. The methodology is adequate and explicitly stated. The quality and quantity of presented data are suitable and completed. The discussion section is completed and very well argued and written. I recommended a minor revision of the manuscript for its improvement before publication.

Some minor comments for the authors:

-The introduction section is completed, and in general the data is up to date (approximately 20% of the references are from 2017-2021).

Line 47-48: insert superscript. Change 105 to 106 by 105 to 106 and 108 to 1010 by 108 to 1010.

-Material and Methods:

In section 2.1, please justify the selection of the different light intensities for the experiment. Are these realistic conditions and do they commonly occur in nature in these organisms?

In the line 149 “of each of the 31 samples were taken with…” refers to 31 samples, however in section 2.1 it describes 8 samples per group (in 4 groups) which would be a total of 32 samples.

-Results:

Line 232: add a full stop in the sentence please.

Round 2

Reviewer 1 Report

The authors have addressed most of the raised points very satisfactorily, and the manuscript has gained in clarity and strength. I would just strongly recommend some edits here in there (see the marked PDF, please attached), in particular in the Results section in the part of "Growth", and in the Discussion, where some arguments are very vague in the parts indicated in the marked PDF.

I hope the authors can address these points soon.

Author Response

Dear Reviewer,

thank you for your second review of our manuscript with the title: “Light availability affects the symbiosis of sponge specific cyanobacteria and the fitness of the Common Blue Aquarium Sponge (Lendenfeldia chondrodes).”

We addressed the remaining concerns and suggestions as follows:

In the Growth section you do not expose any statistical results, p-values, etc... You should have done those comparisons... Even if not significantly discriminative, you should add them, please.

              We added a paragraph with this information in lines 217- 225.

Can you refer to something this percentage? Percentage with respect to original size?

              We added the information:

 ….grew less than 15% with respect to day 0….

in line 211.

Missing space in caption of figure 5.

              We added the missing space.

I would label "Photosynthetic yield" in the X axis.

              We changed the label accordingly.

What do you mean by a "small shading effect"?. Use a better argument related to the physiology of what occurs in Xestospongia to compare your contrasting findings with those.

All this part needs deep editing and focus to expose clear, strong arguments that discuss and explain your dissimilar findings with respect to those of Xestospongia. Surely, defining what you mean by "shading effect" in means of physiology (growth, symbiont densities, pigments, etc... in Xestospongia paper) traits would help to clarify this part.

We rewrote the entire paragraph and discussed in more detail the different findings. We also clarified the ‘shading effects’ and compared them more clearly with our findings.

Something missing here, grammar sounds weird.

              We changed the sentence in line 295 to:

The ecological interpretation of our results needs to be treated with caution since we used clonal sponge samples in our experiment.

I would say "outcomes in our observations".

              We changed the sentence in line 296/297 to:

Any genetic population variability is therefore lacking, which may have affected the outcomes in our observations.

For instance, this is a good argument to discuss differences found with your findings and the paper of Xestospongia. Host related traits causing different symbiotic responses... Just an idea.

              See above. This topic was addressed in the new paragraph mentioned above.

would allow to identify?

              We changed the sentence in lines 391-393 to:

Also, assays investigating carbon sources and flows could identify how resources are allocated within the symbiosis in more detail.
